# Thermal and Structural Characterization of Two Crystalline Polymorphs of Tafamidis Free Acid

**DOI:** 10.3390/molecules27217411

**Published:** 2022-11-01

**Authors:** Norberto Masciocchi, Vincenzo Mirco Abbinante, Marco Zambra, Giuseppe Barreca, Massimo Zampieri

**Affiliations:** 1Dipartimento di Scienza e Alta Tecnologia e To.Sca.Lab., Università dell’Insubria, Via Valleggio 11, 22100 Como, Italy; 2Chemessentia s.r.l, Via Bovio 6, 28100 Novara, Italy

**Keywords:** Tafamidis, polymorphs, crystal structure, powder diffraction, thermal stability

## Abstract

Tafamidis, chemical formula C_14_H_7_Cl_2_NO_3_, is a drug used to delay disease progression in adults suffering from transthyretin amyloidosis, and is marketed worldwide under different tradenames as a free acid or in the form of its meglumine salt. The free acid (CAS no. 594839-88-0) is reported to crystallize as distinct (polymorphic) crystal forms, the thermal stability and structural features of which remained thus far undisclosed. In this paper, we present—by selectively isolating highly pure batches of Tafamidis Form 1 and Tafamidis Form 4—the full characterization of these solids, in terms of crystal structures (determined using state-of-the-art structural powder diffraction methods) and spectroscopic and thermal properties. Beyond conventional thermogravimetric and calorimetric analyses, variable-temperature X-ray diffraction was employed to measure the highly anisotropic response of these (poly)crystalline materials to thermal stimuli and enabled the determination of the linear and volumetric thermal expansion coefficients and of the related indicatrix. Both crystal phases are monoclinic and contain substantially flat and π-π stacked Tafamidis molecules, arranged as centrosymmetric dimers by strong O-H···O bonds; weaker C-H···N contacts give rise, in both polymorphs, to infinite ribbons, which guarantee the substantial stiffness of the crystals in the direction of their elongation. Complete knowledge of the structural models will foster the usage of full-pattern quantitative phase analyses of Tafamidis in drug and polymorphic mixtures, an important aspect in both the forensic and the industrial sectors.

## 1. Introduction

Crystal polymorphism of molecular and covalent compounds has recently permeated the field of solid-state organic chemistry [1] for dyes and pigments [2], organic photovoltaics [3] and, relevant here, in drug and active pharmaceutical ingredient (API) development, processing and formulation [4]. Different and important aspects of basic physico-chemical interest (structure, conformation, stability, reactivity, etc.), in applied science (crystallization, micronization and solvent incorporation/elimination effects) and industrial process control and optimization—up to patent litigation and infringing issues—are thoroughly considered during all these studies. The reason for such flourishing research activities originates from the well-known awareness that different polymorphs can possess significantly different (bio)chemical and rheological properties (hence, different therapeutic activities, tableting and formulation difficulties, shelf lives, etc.). Additionally, when a spontaneous, or poorly controlled, polymorphic change (a solid–solid phase-transition maintaining molecular integrity) occurs, e.g., during formulation and production and/or caused by prolonged storage, the phase purity of the API at the final marketing and delivery stages cannot be firmly controlled, with evident health (and economic) risks. [5]

That APIs are among the most prolific polymorphic family should not surprise, as the discovery of (so) many different solid-state forms (including hydrates, solvates, salts and cocrystals [6]) is definitely governed by the efforts and the amount of time and money spent on their isolation and use.

Once dissolved in body fluids, the individual molecules of an API do not mutually interact and reasonably behave in the same manner no matter which solid form they originate from; at variance, within each polymorph, they possess specific molecular arrangements—dictated by distinct intermolecular interactions and conformationally driven crystal packing—leading to remarkably different physico-chemical properties. In a non-exhaustive list, these include melting points, crystal polarity and chirality, solubility, thermal and chemical stability (or inertness), crystal size and size distribution, crystal shape and preferential cleavage, hardness and compressibility [7]. All these effects can significantly affect their synthesis, isolation and purification processes, as well as several formulation steps, where material rheology influences drug transport, flowability and tableting. Additionally, API’s particle size and shape distributions and its chemical instability (e.g., hygroscopicity) may also affect post-formulation stability over time (e.g., shelf life of the marketed drug). More importantly, bioavailability and pharmacokinetic properties may also be dramatically changed [8], making the drug useless—as for Ritonavir [9,10]—or even toxic [11].

The thermodynamic behavior of different polymorphs can be defined as enantiotropic or monotropic, depending on the topology of their P/T and ΔH, ΔG/T phase diagrams, illustrating their stability ranges [12]. Conventional experimental methods employed to assess, at constant pressure, the relative thermal stability of polymorphs are thermoanalytical (DSC/TGA) or variable-temperature X-ray powder diffraction (VTXRD) analyses. Coupling these two techniques with structural powder diffraction analysis, which can provide atomic-scale models of polymorphs that cannot be isolated as single crystals of suitable size and quality, is now possible, and has largely benefited from the pioneering work of a few academic research groups [13], including ours [14].

In our daily work, we recently faced still-unsolved problems related to the crystal chemistry of Tafamidis, or 2-(3,5-dichlorophenyl)-6-benzoxazolecarboxylic acid (CAS no. 594839-88-0), an ingredient of several marketed drugs (mostly under the tradenames of Vyndamax or Vyndaqel). The chemical structure of the Tafamidis molecule is shown in Figure 1. Approved by the EMA and FDA in 2019, oral formulations of Tafamidis are employed in the treatment of transthyretin (TTR) amyloidosis in adult patients with early-stage symptomatic polyneuropathy to delay peripheral neurological impairment of TTR familial amyloid polyneuropathy.

Tafamidis free acid, in the solid state, has been the subject of several investigations, most of which are inserted in the pertinent patent literature. Suffice to say, several polymorphic forms have been isolated, or claimed, and the only firm structural report is dedicated to Form 2, the structural model of which is not publicly available. Form 1 and Form 4, which are the subject of the present study, have been proposed as pure polymorphs [15], but neither lattice metrics nor, consequently, structural models are presently available. These two forms appear to crystallize, selectively or in combination, during material processing; since the thermodynamics of their possible interconversion are not presently known, we deemed it necessary to study these aspects, additionally motivated by providing structural models making identification and quantification easy, and to interpret their crystal-chemical behavior on a sounder basis. Finally, it must be noted that there exist incomplete reports on a few Tafamidis solvates and on another anhydrous polymorph (showing a complex and uninterpretable XRD powder pattern), which, in this contribution, are not further discussed [16].

Hereafter, we report on the solid-state properties of Tafamidis Form 1 and Form 4, which include spectroscopic FT-IR characterization (a common, fast and cheap analytical method routinely used in the laboratory for the characterization of molecular polymorphs [17]), the less conventional complete structural determination by X-ray powder diffraction methods and thermal characterization by complementary TG, DSC and VTXRD analyses. The results here obtained, and discussed below, clarify the nature of polymorphic transformation of Tafamidis free acid and provide the full structural model which can be fruitfully used in further computational, structural and analytical studies. Therefore, by this study, we filled the still-existing gap between synthetic aspects (the molecular level) and the functional properties (the pharmacological performance) of Tafamidis free acid, adding relevant information on the solid-state properties of the drug.

Worthy of note, in the absence of a single crystal of suitable size and quality for conventional X-ray diffraction structural analysis, other structural methods have recently become viable. These include structural powder diffraction methods [18] (well beyond their common fingerprinting use), synchrotron X-ray single-crystal analysis of tiny specimens (down to 10 μm samples [19,20]) and, more recently, electron diffraction methods, which have been demonstrated to provide approximate, but reliable, structural models for crystalline grains with edges as low as 50–100 nm [21]. In this respect, the much wider accessibility of powder diffractometers than that of synchrotron beamlines or ED accessories for an electron microscope makes the determination of crystal structures of moderately complex molecular compounds (e.g., drugs) a viable option, at the expenses of less accurate models [22]. Accordingly, this contribution sagaciously uses the powder diffraction method in the complementary structural and thermodiffractometric modes, and opens the way to the use of XRD-based quantitative phase analysis of polymorphic mixtures [23], a problem which permeates the academic, and more relevantly, the forensic [24] and industrial [25], sectors.

## 2. Materials and Methods

### 2.1. Materials

A large batch (>100 g) of chemically pure solid Tafamidis of unknown crystal form was supplied by Química Sintética S.A., Alcalá de Henares, Spain. Crystallographically pure Tafamidis Form 1 and Tafamidis Form 4 were prepared according to the synthetic procedures described in patent WO 2016/038500 Al, in ca. 10 g batches [16]. Their phase purity was assessed by XRD after successful indexing and structure solutions were attained (*vide infra*).

### 2.2. Methods

#### 2.2.1. IR Spectroscopy

Spectral data were obtained with FTIR equipment (Nicolet iS10 FTIR Spectrometer, Thermo Scientific, Rodano (Milano, Italy) using a diamond single-reflection attenuated total reflectance (ATR) device. Spectra were collected using 64 scans at 1 cm^−1^ resolution in the spectral interval range of 4000 to 650 cm^−1^.

#### 2.2.2. Thermal Analyses

Thermogravimetric (TG) traces were acquired from 30 to 400 °C (with a scan rate of 10 °C min^−1^) using a Netzsch STA 409 PC Luxx^®^ analyzer under an N2 flow and with alumina sample-holders equipped with a pierced lid. Differential scanning calorimetry (DSC) traces were acquired from 30 to 300 °C (with a scan rate of 10 °C min^−1^) using a Mettler DSC 1 Stare System analyzer under an N2 flow (80 mL min^−1^) and with 40 µL crimped aluminum sample-holders equipped with a pierced lid.

#### 2.2.3. Structural X-ray Powder Diffraction Analysis

Tafamidis Form 1 and Form 4 were gently ground in an agate mortar and then deposited in the hollow of a 0.2 mm deep silicon monocrystal (a zero background plate, supplied by Assing spa, Monterotondo, Italy). Diffraction data were collected in the 5–105° 2θ range on a D8 Bruker AXS vertical sampling diffractometer operating in θ:θ mode, equipped with a linear Lynxeye position-sensitive detector, set at 300 mm from the sample (Generator settings: 40 kV, 40 mA, Ni-filtered Cu-Kα_1,2_ radiation, λ_avg_ = 1.5418 Å). Peak search and profile-fitting allowed for the location of the most prominent, low-angle peaks, which were later used in the indexing process by the TOPAS-R software [26]. Approximate lattice parameters of primitive monoclinic cells were determined to be a = 22.99, b = 14.99, c = 3.79 Å and β = 91.0° [GOF(29) = 23.8] for Form 1 and a = 22.33, b = 15.18, c = 3.72 Å and β = 93.9° [GOF(27) = 20.4] for Form 4. Systematic absence conditions suggested *P2_1_/a* (Form 1) and *P2_1_/n* (Form 4) as the probable space groups, later confirmed by successful structure solution and refinement. The choice of maintaining non-standard, but tolerated, space-group settings is here motivated by having comparable crystal cell axis lengths and orientations. Note that the occurrence of a rather short **c** axis in both forms required relaxation during the solution process, as **c**—as well as the β angle—are determined with low accuracy (as per the well-known dominant zone problem). Density considerations indicated Z = 4, thus limiting the structural solution process of both forms to the individuation of the center-of-mass location, orientation and some conformational freedom of an otherwise-rigid Tafamidis molecule, defined by z-matrix formalism. Real-space structure modeling by the simulated annealing algorithm, coupled with a Monte Carlo search, allowed the definition of suitable models, which were later refined by the Rietveld method. Structure solution and final refinements were carried out with TOPAS-R software. The background contribution was modelled by a Chebyshev polynomial fit; atomic scattering factors for neutral atoms were taken from the internal library of TOPAS-R. Preferred orientation corrections, in the March–Dollase formulation [27], were applied on the [001] and [301] poles (both bearing an evident structural meaning, vide infra), with final magnitudes g_001_ = 1.261 (2) and g_301_ = 1.087 (2), respectively. Anisotropic peak broadening, modeled by the spherical harmonics approach, led to a significantly lower agreement factor and was, therefore, adopted. Crystal data and relevant data analysis parameters are collected in Table 1. Figure 2 shows the final Rietveld refinement plots. Fractional atomic coordinates have been deposited as CIF files within the Cambridge Crystallographic Database as publications No. CCDC 2209986-2209987.

#### 2.2.4. Variable-Temperature X-ray Powder Diffractometry

Thermodiffractometric experiments were performed from 30 to 300 °C, or slightly beyond, to assess thermal and polymorphic stability. Powdered batches were deposited in the hollow of an aluminum sample-holder of a custom-made heating stage (Officina Elettrotecnica di Tenno, Ponte Arche, Italy). Diffractograms were acquired in air, in the most significant (low-angle) 5–32° 2θ range, under isothermal conditions in 20 °C steps. Since the samples in powder diffraction experiments are in direct contact with the air, and some thermal drifts/gradients are present, accurate transition temperatures are calibrated on TG/DSC measurements. Computation of the thermal strain was performed using Ohashi’s method [28] as implemented by the Bilbao Crystallographic Server [29], assisted by tensor visualization [30], after extracting T-dependent cell parameters by the structureless Le Bail technique [31].

## 3. Results

### 3.1. Comparative Crystal Chemistry

Crystals of Tafamidis Form 1 and Form 4 are both monoclinic, and share space-group symmetry (standard *P2_1_/c*, N. 14). However, in order to simplify the crystal-chemical comparison, their structures are here described in non-standard settings (*P2_1_/a* and *P2_1_/n*, respectively), which keep the length of the **c** axes nearly unchanged, as they are strictly related to the stacking sequence of nearly planar molecules.

Laboratory X-ray powder diffraction studies of organic molecular systems of moderate complexity do not provide robust information on individual bond-distances and angles. At variance, since the final structural model is known to be more sensitive to molecular packing and conformational aspects, the only structural parameters that are relevant for the stereochemical discussion are the molecular τ_1_ and τ_2_ torsional angles (depicted in Figure 1) and ancillary geometrical entities, synoptically collected in Table 2. Figure 3 shows the different molecular and packing features for Tafamidis Form 1 and Tafamidis Form 4. Jointly with the values reported in Table 2, it is evident that both phases contain Tafamidis molecules in similar (though statistically distinct) conformations (not easily detectable at the drawing level, see Figure 3a,b). Through inversion centers, individual molecules give rise to hydrogen-bonded dimers, typical of monocarboxylic acids, with very similar geometries (O–H···O = 2.62–2.64 Å). A further supramolecular aspect is the presence, in both polymorphic phases, of infinite ribbons running along **b** (later discussed), generated by the alternate sequence of carboxylic dimers and centrosymmetrically related (definitely weaker) C–H···N interactions linking two oxazole rings across an inversion center (3.46–3.57 Å). This feature also explains the similarity of the **b** axis length in Tafamidis Form 1 and Form 4.

### 3.2. High-Temperature Diffraction Studies

Variable-temperature X-ray diffraction was used to assess the thermal evolution of Tafamidis Form 1 and Form 4, in the form of polycrystalline powders, when heated in air from room temperature to material sublimation. The 2D thermodiffractograms, shown in Figure 4a and Figure 5a, clearly manifest the substantial constancy of the crystal phase of Form 4. Differently, when Form 1 is progressively heated, a change of peak positions and intensities can be observed near 240 °C and can be easily explained by the quantitative formation of Form 4 through a solid–solid polymorphic transformation.

Using the angular variation of the peak position upon heating, a significant anisotropy of the thermal-expansion tensor was derived, the isosurface plot of which is drawn in panels b of Figure 4 and Figure 5. As expected, both isosurfaces show a prolate form pointing nearly toward z (i.e., the **c** axis). Such behavior is not unexpected, given that we deliberately chose the cell axis orientation (with the adoption of two non-standard settings of the *P2_1_/c* space group) in such a way that axes were “nearly” orthogonal, and the sequence and lengths of *all* axes were comparable. The thermal expansion values, computed in the 25–125 °C range (ΔT = 100 K) using the formula x_125_ = x_25_ (1 + 100κ_x_), with x = a, b, c, β, V, are reported in Table 3. They highlight a significant stiffness in the *xy* plane and, particularly, along the **b** axis. The presence of supramolecular contacts generating infinite molecular ribbons, with the hinges highlighted in the yellow dashed circles in Figure 4 (for Tafamidis Form 1), is considered responsible for such thermal inertness. A nearly identical situation is present in Tafamidis Form 4 (not shown here).

### 3.3. Thermal Analyses

The thermal behavior of Tafamidis Form 1 and Form 4 were also studied by thermogravimetric and differential calorimetric analyses. Their TG and DSC traces are shown in Figure 6, and manifest a substantial chemical inertness, with material sublimation occurring only above 280 °C. The DSC curves, however, are much more informative, and provide enthalpic values in line with the occurrence of a solid-state polymorphic transformation for Form 1 (onset at 226 °C), with minimal heat absorption (ΔH = 0.8 kJ mol^−1^) and the formation of Tafamidis Form 4. This latter form melts above 280 °C (onset at 283 °C) with a latent heat of fusion of 40.6 kJ mol^−1^ (see Figure 6a). As expected, the DSC curve of Form 4 is much flatter, and melting occurs with a ΔH_fus_ of 39.5 kJ mol^−1^, in reasonable agreement with the previously quoted value (see Figure 6b).

Significantly, sublimation occurs without material decomposition. Indeed, a crystalline material could be recovered upon collecting powders sublimed in air at 325 °C on a metal blade kept at room temperature ca. 20 mm above the heating stage. Through conventional qualitative XRD fingerprinting and quantitative XRD phase analysis (enabled by the full knowledge of the structural models), such white material, of cotton-like appearance, proved to be a biphasic mixture with an approximate relative content of 13 and 87 *w*/*w* % Form 1 and Form 4, respectively.

The endothermic transformation of Form 1 into Form 4 and the subsequent melting of the latter indicate that Form 1 (at low enough temperatures) is more stable than Form 4, and that the two forms are enantiotropically related. However, Tafamidis Form 4—which should restore to Form 1 upon cooling—was found to be indefinitely stable (for months), as if a too-high energy barrier existed for the inverse transformation, which cannot be active once the temperature is (more or less) abruptly lowered.

A further experiment was also performed, aiming at proving the relative stabilities of Form 1 and Form 4. Upon heating powders of severely ground Tafamidis Form 4 on a hot metal plate set at 280 °C, progressive crystal growth of tiny needles on the cool side in contact with environmental air occurred, with the formation of a relatively homogeneous mat of intertwined crystals. This material was then characterized by quantitative XRD analysis, and proved to be nearly pure Form 1 (residual Form 4 accounted for ca. 2% *w*/*w*). Based on this result, we can safely state that if sublimation/recrystallization occurs *slowly* at high enough temperatures, the Form 1 → Form 4 phase transformation can be reverted; such observation is fully in line with the enantiotropic polymorphic relationship claimed above.

In addition, in contrast with the commonly accepted Kitaigorodskii [32] and Burger and Ramberger [33] rules, where the denser polymorph is normally considered the more stable form, here Form 1 is less dense than Form 4 (see Table 1). This aspect, however, is not worrisome, as this rule is often broken in hydrogen-bonded molecular crystals [34], as in the present case and in the widely marketed drug paracetamol [35].

### 3.4. IR Fingerprinting

One of the fastest, easiest and cheapest analytical methods used in API characterization and quality control is undoubtedly FTIR spectroscopy, particularly in the Attenuated Total Reflection (ATR-FTIR) mode [36]. To prove, or dismiss, the power of this technique when applied to Tafamidis polymorphs, we collected ATR-FTIR data which, in the most significant region, are graphically compared in Figure 7. In more detail, the lists of transmittance signals are here provided, with Tafamidis Form 4 values set in parentheses: 1688s (1692), 1613w (1619), 1589w (1589), 1572w (1572), 1546m (1546), 1493w (1492), 1438m (1438), 1423m (1424), 1410m (1410), 1388w (1386), 1349w (1349), 1309m (1309), 1291s (1290), 1272s (1274), 1236w (1237), 1196w (1194), 1134w (1134), 1129w (1126), 1102w (1103), 1087w (1086), 945w (944), 935w (935), 886m (887), 860s (859), 848w (850), 807m (807), 790m (792), 773s (773), 745m (745), 726s (726) and 665s (665).

This extensive list and the graphs in Figure 7 clearly demonstrate that, for this specific case, state-of-the-art ATR-FTIR cannot distinguish between the two polymorphs, as if their vibrational patterns were completely dominated by molecular modes without any contribution from intermolecular contacts. Indeed, the differences between the corresponding peaks are limited to a few cm^−1^ or even less, i.e., close to the resolution limit of any standard IR spectrometer. As our XRD study clearly proved that the basic building units of the entire crystals in both polymorphs (the Tafamidis centrosymmetric dimers) are practically identical, and that no evident “strong” interactions connect the different dimers, this result should not surprise. In such cases, Raman spectroscopy in the low-frequency range is claimed to be much more informative [37].

## 4. Conclusions

In this paper, we determined fundamental properties of a widely marketed drug (Tafamidis free acid), in the two known (polycrystalline) polymorphic Form 1 and Form 4 phases. Our study closes the existing gap between the molecular and the pharmacological levels, and provides (otherwise inaccessible) structural information through the sagacious usage of structural X-ray powder diffraction methods. Such detailed knowledge, which encompasses crystal symmetry, lattice parameter and fractional coordinates of all atoms in the structure, is indeed crucial for the identification of contaminants and enables the reliable relative quantification of the two forms in polyphasic mixtures by state-of-the-art whole-profile (Rietveld-like) quantitative analysis [38].

Additionally, we coupled common TGA/DSC methods and the less conventional thermodiffractometry, and studied the thermodynamic landscape of the polymorphic solid–solid transformation and its irreversible nature when cooling is (more or less) rapidly executed. A complete thermal strain tensor analysis was also performed, and the size and shape of the thermal strain indicatrix was determined. Finally, we proved that, in this case, the extreme similarity of the FTIR transmittance spectra—typically used for cheap and fast polymorphic recognition and fingerprinting—makes this analytical method not viable, and that resorting to XRPD becomes necessary.

## Figures and Tables

**Figure 1 molecules-27-07411-f001:**
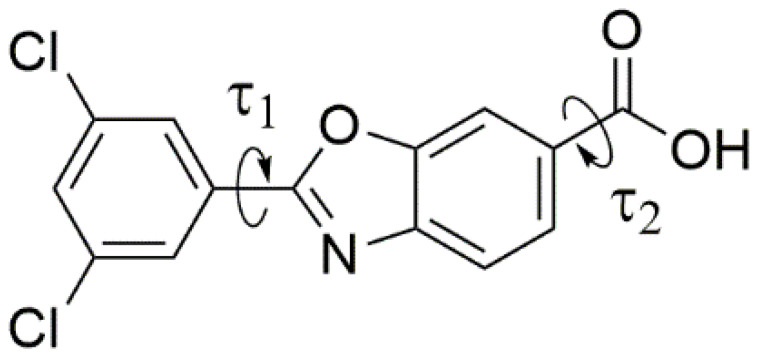
The molecular scheme and labeling of the freed torsion angles of the title compound.

**Figure 2 molecules-27-07411-f002:**
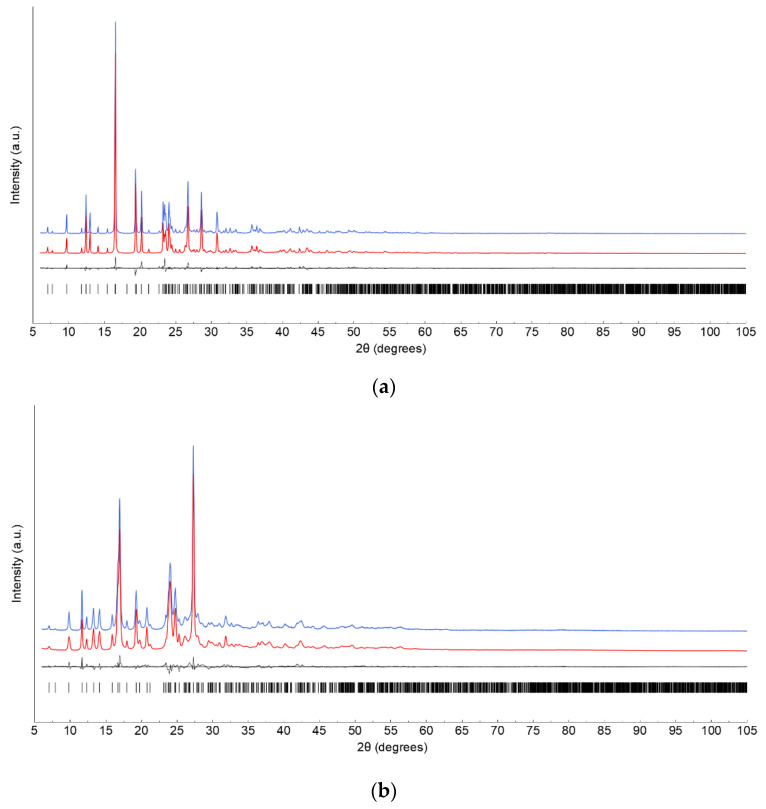
Rietveld refinement plots for Tafamidis Form 1 (**a**) and Tafamidis Form 4 (**b**), with peak markers and difference plot at the bottom. Observed data in blue and calculated trace in red, offset on the y axis for clarity.

**Figure 3 molecules-27-07411-f003:**
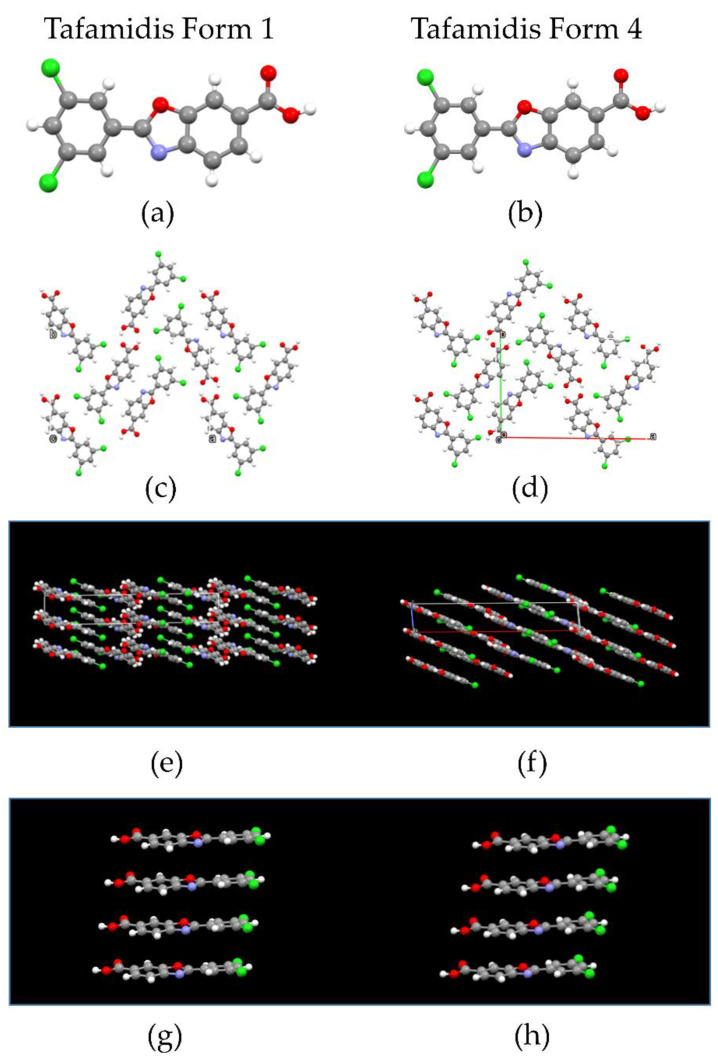
Comparative drawing of the most relevant molecular and packing entities of Tafamidis Forms 1 (left column) and 4 (right column) (**a**,**b**) the individual molecules; (**c**,**d**) the crystal packing viewed down **c**; (**e**,**f**) the crystal packing viewed down b, where the largest differences are visible; (**g**,**h**) the π-π stacking occurring along **c**, characterized by significantly different interplanar vectors: 3.51 and 3.41 Å, respectively. Color codes: C (grey); H (white), N (blue), O (red) and Cl (green).

**Figure 4 molecules-27-07411-f004:**
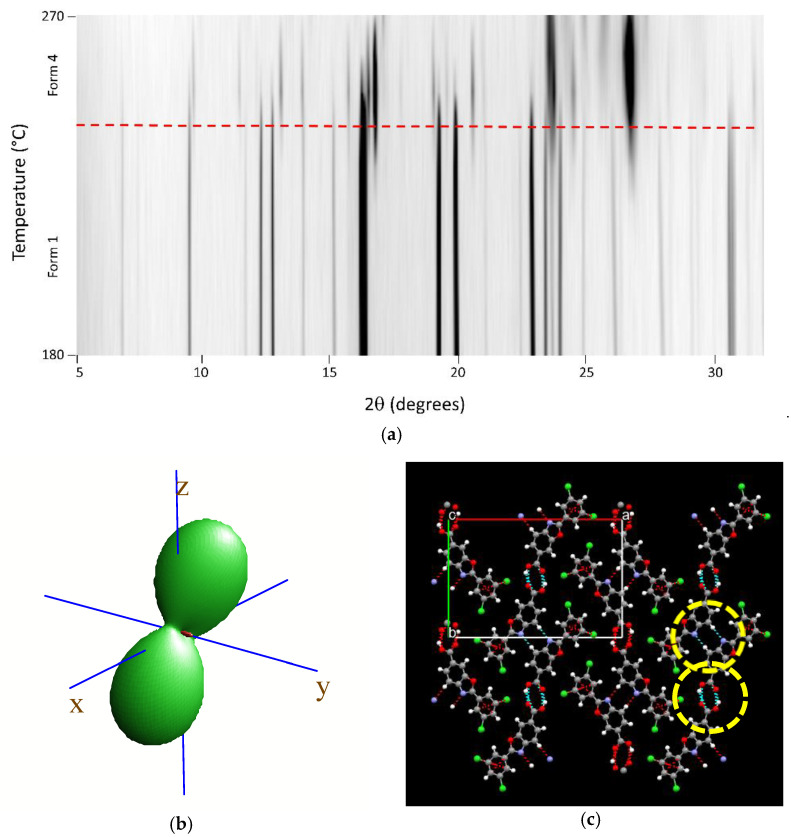
Variable-temperature X-ray diffraction (VTXRD) data (in the most relevant 180–270 °C range) showing the thermal evolution of the XRD pattern upon heating Tafamidis Form 1 in situ in the diffractometer cradle (**a**), showing its quantitative conversion into Form 4 at ca. 240 °C (red dashed line). (**b**) Visualization of the thermal strain tensor, showing nearly null (actually, slightly negative) thermal expansion in the y direction (see text), and (**c**), the hydrogen-bonded ribbons running along **b**. Indeed, couples of short O-H···O and C-H···N contacts, highlighted by the yellow circles, are responsible for the stiffness of the structure along **b**. Note that, for monoclinic symmetry, the y direction in the plot of the tensor isosurface is aligned with **b,** and that the *xz* plane corresponds to the crystallographic **ac** plane. Color codes in panel (**c**): C (grey); H (white), N (blue), O (red) and Cl (green).

**Figure 5 molecules-27-07411-f005:**
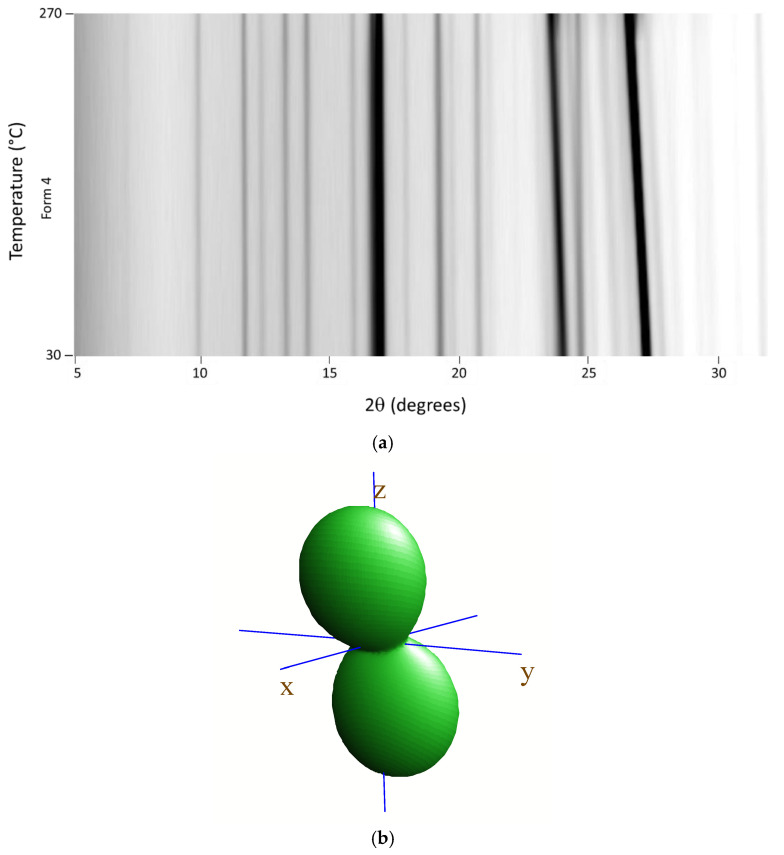
Variable-temperature X-ray diffraction (VTXRD) data (in the full 30–270° range) showing the thermal evolution of the XRD pattern upon heating Tafamidis Form 4 in situ in the diffractometer cradle (**a**); note that this form does not suffer from any phase transformation before melting. (**b**) Visualization of the thermal strain tensor, showing a relatively large thermal expansion in the z direction and much smaller (but still positive) components in the *xy* plane (see text). Note that, for monoclinic symmetry, the y direction in the plot of the tensor isosurface is aligned with **b,** and that the *xz* plane corresponds to the crystallographic **ac** plane.

**Figure 6 molecules-27-07411-f006:**
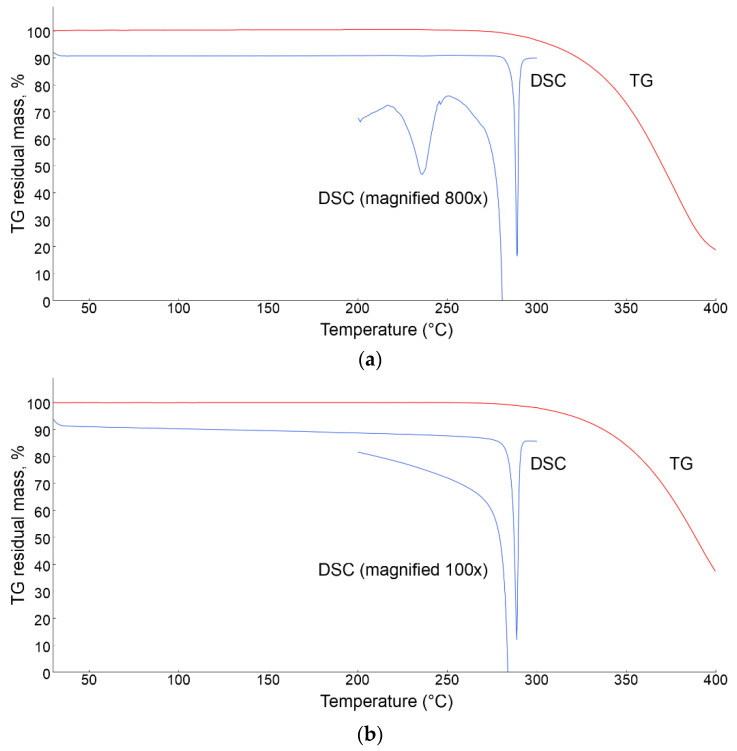
Thermodiffractometric (TG, in red) and calorimetric (DSC, in blue) curves for Tafamidis Form 1 (**a**) and Tafamidis Form 4 (**b**). The magnification of the DSC trace of panel (**a**) shows a broad endothermic event peaking near 240 °C, related to the Form 1 to Form 4 transformation. This peak is absent in Form 4, while melting of Form 4 (onset at ca. 283 °C) is observed in both samples.

**Figure 7 molecules-27-07411-f007:**
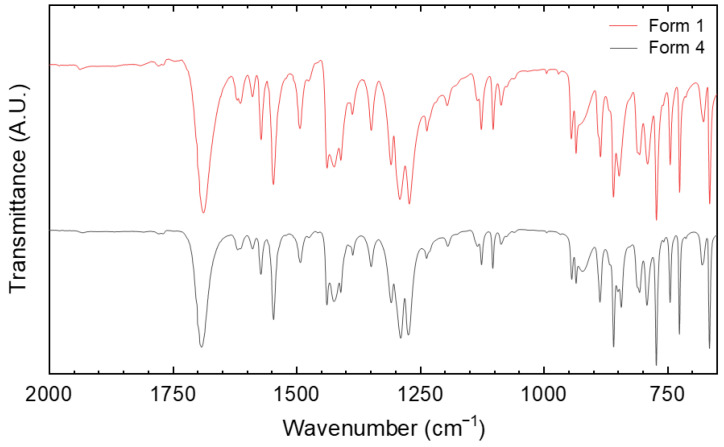
Plot of the ATR-FTIR transmittance spectra, in the most relevant portion (2000–650 cm^−1^ range), clearly showing the substantial identity of the FTIR traces of Tafamidis Form 1 and Form 4, preventing polymorph recognition by conventional vibrational spectroscopy (see text).

**Table 1 molecules-27-07411-t001:** Crystal data and data analysis parameters for Tafamidis Form **1** and Form **4**.

Parameter	Form 1	Form 4
Formula	C_14_H_7_Cl_2_NO_3_	C_14_H_7_Cl_2_NO_3_
fw, g mol^−1^	308.12	308.12
Crystal system	monoclinic	monoclinic
Space group	*P2_1_/a* (No. 14)	*P2_1_/n* (No. 14)
a, Å	22.976 (1)	22.364 (3)
b, Å	14.993 (1)	15.174 (2)
c, Å	3.794 (1)	3.819 (1)
β, °	90.938 (3)	95.265 (5)
V, Å^3^	1306.9 (1)	1290.7 (3)
Z	4	4
V/Z, Å^3^	326.7	322.27
ρ_calc_, g cm^−3^	1.566	1.586
μ (CuKα), cm^−1^	45.5	46.1
F (000)	624	624
λ_avg_, Å	1.5418	1.5418
T, K	295	295
2θ range, °	6–105	6–105
R_p_, R_wp_	0.079, 0.103	0.062, 0.083
χ^2^	4.13	5.16
R_Bragg_	0.064	0.039

**Table 2 molecules-27-07411-t002:** Comparison of relevant stereochemical and geometrical parameters highlighting similarities and differences between the two polymorphic forms of Tafamidis.

	Form 1	Form 4	A Sketch of the SV, χ and ψ Parameters
τ_1_ torsional angle, °	9.4 (0.2)	1.5 (0.3)	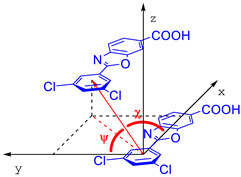
τ_2_ torsional angle, °	0.0 (0.5)	6.0 (0.5)	
O–H···O, Å	2.62	2.64	
Stacking Vector (SV), Å	3.794	3.819	
χ angle, °	78.8	70.9	
ψ angle, °	80.9	71.8	
Interplanar Distance, Å	3.51	3.41	

**Table 3 molecules-27-07411-t003:** Comparison of linear and volumetric thermal expansion coefficients, calculated in the 25–125 °C range, using the formula x_125_ = x_25_ (1 + 100κ_x_), with x = a, b, c, β, V.

	Form 1	Form 4
κ_a_, 10^6^ K^−1^	50	20
κ_b_, 10^6^ K^−1^	−14	22
κ_c_, 10^6^ K^−1^	74	93
κ_β_, 10^6^ K^−1^	−48	−29
κ_V_, 10^6^ K^−1^	108	141

## Data Availability

Data are available from the authors upon request.

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
