# Peer review of "Thermal and Structural Characterization of Two Crystalline Polymorphs of Tafamidis Free Acid"

_molecules, 2022, doi:10.3390/molecules27217411_

Round 1

Reviewer 1 Report

This work “Solid-state characterization of crystalline polymorphs of Tafamidis”, the authors consider the, Tafamidis Form 1 and Tafamidis Form 4 were selectively prepared and characterized by thermal, spectroscopic and state-of-the-art structural powder diffraction methods, which disclosed the different packing features of substantially flat and pi-pi stacked Tafamidis molecules, arranged as centrosymmetric dimers, in the two forms.

Although the topic is interesting, the authors have not provided novelty in a significant way and the presentation of the manuscript should be improved in a significant way too.

·        

·       The title of the manuscript can be modified for better understanding.

·       The abstract of the article is very confusing and must be restructured for a better understanding of the readers specifically about the methoddolgy used. Secondly, results are not clearly defined in the second last sentence of the abstract. The abstract is not defining any problem statement, it is explaining just introduction.

·       There are many grammatical errors throughout the paper, which need to be corrected. Try to avoid unnecessary long sentences.

·        The literature review is ambiguous; include some more recent state-of-the-art papers in the Literature review for better understanding which are more relevant.

·       The Introduction part of the article must be revised to make it better structured for the readers. Try to explain the previous work related to different aspects of the current research and connect it with the problem statement in the end i.e. identifying the gap and why was this model necessary to develop. An intense revision is required in this section.

·       In the last paragraph of the introduction section, mention the novelty of this paper with the previous state-of-the-art research in a better way. Moreover, mention the applications of this work.

·       The novelty of the manuscript is not very significant. Try to improve it.

·       Results are poorly written. Try to explain it in a better way with all the considerations and physics behind it.

·       Conclusions should be restructured and more concrete from mathematical point.

Author Response

This work “Solid-state characterization of crystalline polymorphs of Tafamidis”, the authors consider the, Tafamidis Form 1 and Tafamidis Form 4 were selectively prepared and characterized by thermal, spectroscopic and state-of-the-art structural powder diffraction methods, which disclosed the different packing features of substantially flat and pi-pi stacked Tafamidis molecules, arranged as centrosymmetric dimers, in the two forms.

Although the topic is interesting, the authors have not provided novelty in a significant way and the presentation of the manuscript should be improved in a significant way too.

“We thank the reviewer for his partially positive comments and for the criticisms which helped us to assemble a more intelligible version, now proposed with several modifications and additions. In the following we address mort of the points raised by this reviewer, taking also into account that we substantially, but not completely, modified our manuscript, which received two very positive reviews from the other referees.”

The title of the manuscript can be modified for better understanding.

“The title has been rephrased, giving a specific focus to the subject of this study. It now reads as:
Thermal and Structural Characterization of Two Crystalline Polymorphs of Tafamidis Free Acid”

The abstract of the article is very confusing and must be restructured for a better understanding of the readers specifically about the methodology used. Secondly, results are not clearly defined in the second last sentence of the abstract. The abstract is not defining any problem statement, it is explaining just introduction.

“The abstract of the article has been substantially revised, aiming at making the readers aware of the methods used and of the results of our structural and thermodiffractometric study. These modifications also address one concern of another reviewer.”

There are many grammatical errors throughout the paper, which need to be corrected. Try to avoid unnecessary long sentences.

“Through a careful text revision, we have corrected several typo’s. Additionally, to better convey the results of our studies, we have also rephrased a few long sentences, either by shortening or splitting them.”

The literature review is ambiguous; include some more recent state-of-the-art papers in the Literature review for better understanding which are more relevant.

“In order to cope with the reviewer request, we have included several additional (and more recent) papers in the list of references, helping the interest readers to expand, at their will, their curiosity in the different fields and aspects. However, we still do not understand why the original literature review was considered “ambiguous”. Perhaps other adjectives, such as “incomplete”, “biased”, “outdated” may have better conveyed the legitimate opinions of the referee, but “ambiguous” makes, in our humble opinion, little sense.”

The Introduction part of the article must be revised to make it better structured for the readers. Try to explain the previous work related to different aspects of the current research and connect it with the problem statement in the end i.e. identifying the gap and why was this model necessary to develop. An intense revision is required in this section. In the last paragraph of the introduction section, mention the novelty of this paper with the previous state-of-the-art research in a better way. Moreover, mention the applications of this work.

“The introductory section, which did not receive any (positive or negative) comments from the other reviewers, has been modified by adding, in the last section, two long paragraphs, with the insertion of several new references. In particular, in the revised version, we explicitly mentioned the state-of-the-art in structural and polymorphic characterization of drugs, and the direct fall-out guaranteed by the knowledge of the full list of atomic coordinates, in the forensic and industrial sectors, to mention a few.”

The novelty of the manuscript is not very significant. Try to improve it.

“As now explicitly mentioned in the text (in the Introduction and in the Conclusion sections), the novelty of this contribution can be summarized by the thorough characterization of both Form 1 and Form 4 of Tafamidis free acid, which fills the still existing gap between synthetic aspects (the molecular level) and the functional properties (the pharmacological performances).”

Results are poorly written. Try to explain it in a better way with all the considerations and physics behind it.

“This part has been only marginally changed, since we (and the other reviewers) did find it understandable. Nevertheless, the legitimate concern of this referee should find satisfactory answers in the several (much more extensive) modifications inserted in the Introduction, in the Conclusion and in the newly added references.”

Conclusions should be restructured and more concrete from mathematical point.

“This section has been shortened, aiming at clarifying the most important findings. We reshaped this section into more focused statements, as indeed suggested by the reviewer, making the Conclusions more direct and separating them into two well distinct portions.”

Reviewer 2 Report

In this manuscript, the authors report the two forms of Tafamidis and study the thermal, spectroscopic and state-of-the-art structural powder diffraction. Overall, this paper is well written and the data is convincing and I recommend its acceptance to Molecules. Before publication, the following minor points should be made.

1.     The color note for each atom in the two forms of Tafamidis should be given in related Figures of main text.

2.     Only the Rietveld Refinement plots for Tafamidis Form 1 and Form 4 are given in Figure 2. More detail data for the results of Rietveld Refinement, like R, GOOF values, etc., should be provided.

3.     The author gives the linear and volumetric thermal expansion coefficients of reported Tafamidis, the direction for x = a, b, c, and β, should be given.

Author Response

In this manuscript, the authors report the two forms of Tafamidis and study the thermal, spectroscopic and state-of-the-art structural powder diffraction. Overall, this paper is well written and the data is convincing and I recommend its acceptance to Molecules. Before publication, the following minor points should be made.

“We thank the reviewer for his/her substantially positive comments on our manuscript.”

  1. The color note for each atom in the two forms of Tafamidis should be given in related Figures of main text.

“The color codes have been added in the captions to Figure 3 and Figure 4.”

  1. Only the Rietveld Refinement plots for Tafamidis Form 1 and Form 4 are given in Figure 2. More detail data for the results of Rietveld Refinement, like R, GOOF values, etc., should be provided.

“Most of data required by the reviewer were already present in the original version in Table 1 (Rwp, Rp, RBragg). Chi-square (the goodness of fit for powder diffraction data – not to be confused with the one normally computed from conventional single crystal analysis) has now been inserted as an extra line in Table 1.”

  1. The author gives the linear and volumetric thermal expansion coefficients of reported Tafamidis, the direction for x = a, b, c, and β, should be given.

“A note on the relationship between crystal and tensor isosurface axes (and planes) has been inserted in the captions to Figure 4 and to Figure 5.”

Reviewer 3 Report

In this paper, Tafamidis Form 1 and Form 4 were prepared and charazterized by different methods. The data is meaningful and the conclusion is important. The suggestions to improve the quality are as following:

1) The title is too general, without any focus on the main innovation of this paper. 

2) Most part of the abstract is talking about the background, other than the cotent of this paper, which is not good. 

3) The conclusions part didn't provide the most important conclusions of this paper.

4) The references are not in uniform format.

Author Response

In this paper, Tafamidis Form 1 and Form 4 were prepared and characterized by different methods. The data is meaningful and the conclusion is important. The suggestions to improve the quality are as following:

“We thank the reviewer for his/her substantially positive comments on our manuscript.”

1) The title is too general, without any focus on the main innovation of this paper.

“The title has been rephrased, given the specific focus to the subject of this study. It now reads as:
Thermal and Structural Characterization of Two Crystalline Polymorphs of Tafamidis Free Acid”

2) Most part of the abstract is talking about the background, other than the content of this paper, which is not good.

“The abstract of the article has been substantially revised, aiming at making the readers aware of the methods used and of the results of our structural and thermodiffractometric study. These modifications also address the concerns of another reviewer.”

3) The conclusions part didn't provide the most important conclusions of this paper.

“This section has been shortened and restructured, aiming at clarifying the most important findings of this study. The Conclusions are now” more direct”, after separating our findings into two well distinct portions.”

4) The references are not in uniform format.

“We have double-checked all references and corrected a few inconsistencies. In the revised version they are presented in a uniform manner. For sake of completeness, we mention that the modifications and additions of new portions in the text required the insertion of new references, inserted with the correct format.”

Round 2

Reviewer 1 Report

authors have answered all the questions of reviewers. Paper can be published in current form.